# HPLC Analysis of Polyphenols Derived from Hungarian Aszú from Tokaj Wine Region and Its Effect on Inflammation in an In Vitro Model System of Endothelial Cells

**DOI:** 10.3390/ijms24076124

**Published:** 2023-03-24

**Authors:** Arnold Markovics, László Csige, Erzsébet Szőllősi, Hajnalka Matyi, Andrea Diána Lukács, Nóra Réka Perez, Zsófia Réka Bacsó, László Stündl, Judit Remenyik, Attila Biró

**Affiliations:** 1Institute of Food Technology, Faculty of Agricultural and Food Sciences and Environmental Management, University of Debrecen, H-4032 Debrecen, Hungary; 2Research Laboratory and Wine Academy of Mad, University of Debrecen, H-3909 Mád, Hungary

**Keywords:** wine, inflammation, endothelial cells, cytokines, flavonoids

## Abstract

Many studies have been published in recent years regarding the fact that moderate wine consumption, as a part of a balanced diet can have a beneficial effect on human health. The biologically active components of wine continue to be the subject of intense research today. In this study, the bioactive molecules of Hungarian aszú from the Tokaj wine region were analyzed using high-performance liquid chromatography (HPLC) and investigated in an in vitro model system of endothelial cells induced by bacterial-derived lipopolysaccharide. The HPLC measurements were performed on a reversed phased column with gradient elution. The non-cytotoxic concentration of the active substance was determined based on 3-[4,5-dimethylthiazol-2-yl]-2,5 diphenyltetrazolium bromide (MTT)-, apoptosis, and necrosis assays. The antioxidant effect of the extract was determined by evaluating its ability to eliminate ROS. The expressions of the interleukin-(IL)1α, IL1-β, IL-6, and IL-8 pro-inflammatory cytokines and nitric oxide synthase (eNOS) at the mRNA level were evaluated using a quantitative polymerase chain reaction (qPCR). We found that the lipopolysaccharides (LPS)-induced increases in the expressions of the investigated cytokines were significantly suppressed by Hungarian aszú extract, excluding IL-6. In our experimental setup, our treatment had a positive effect on the eNOS expression, which was impaired as a result of the inflammatory manipulation. In our experimental model, the Hungarian aszú extract decreased the LPS-induced increases in the expression of the investigated cytokines and eNOS at the mRNA level, which presumably had a positive effect on the endothelial dysfunction caused by inflammation due to its strong antioxidant and anti-inflammatory effects. Collectively, this research contributes to a more thorough understanding of the bioactive molecules of aszú from the Tokaj wine region.

## 1. Introduction

For many centuries, wine has been considered not only an article of pleasure, but also medicine and, in various situations, it was already used as such in ancient cultures [1]. This fact has drawn attention to the potential health benefits of wine ingredients centuries ago. The explanation of the phenomenon referred to as the “French paradox” greatly contributed to the popularization of the idea that wine consumption has a positive effect on health [2].

The potential physiological effects of some of components of wine are continue to be the subject of intensive research today [3,4]. Several clinical and epidemiological studies have proved the positive effects of wine on health and disease prevention, specifically its phytochemical components, such as polyphenols and flavonoids [5,6]. Compared to other food where polyphenol compounds are found in polymer, insoluble or strongly bound form, the advantage of wine is that the polyphenols are present in dissolved form, so their absorption is facilitated [7].

Apostolidou et al. [8] investigated the effects of moderate wine consumption in 40 healthy participants with high cholesterol. In this crossover study, individuals consumed either red wine or a placebo drink (125 mL for women and 250 mL for men daily) for one month. Wine consumption positively influenced the antioxidant capacity and vitamin E levels of both patients with high cholesterol and patients with normal cholesterol. Moreover, the fasting LDL/HDL ratio was also improved in the case of patients with high cholesterol. In a randomized trial, the effects of red and white wine on atherosclerosis were investigated. In total, 157 healthy participants received either white or red wine for one year. After the trial, the LDL decreased in both groups [9]. A cross-sectional study with 5801 elderly people at high cardiovascular risk proved the protective effect of red wine consumption. [10]. A recently published comprehensive review examined studies published in the PubMed database between 2013 and 2018, which investigate the relationship between moderate wine consumption and health. It concluded that moderate wine consumption, 1–2 glasses per day, as part of the Mediterranean diet, is positively associated with promoting human health and preventing disease [11]. Of course, it should be mentioned that binge alcohol consumption has also been associated with negative health impacts [12]. Therefore, the positive effects of wine consumption are to be treated with reservation.

The positive biological effect of wine is attributed to its phytochemical components, which are preserved by the wine even after the fermentation process, such as polysaccharides and acids, and phenolic compounds such as flavonoids and non-flavonoids. Despite many proven health benefits, a lot of discussions are still taking place regarding the actions of the components of wine on cells and molecular interactions [13]. It is important to note that the chemical composition of each wine sample can vary greatly depending on the technology, grape variety and microbial fermentation [14]. The wine sample included in our investigation is aszú from the Tokaj region. The basis of the aszú wine is the optimally botrytized berry (locally called aszú berries), which are brown, with violet hues, raisin-like and fully shrivelled. The winemaking process involves the selective harvest and storage of the noble-rotted berries; producing a must or base wine from sound grapes of the same vintage; then soaking and macerating the botrytized fruit in this fermenting must or wine [15].

In our current research, we focused on the polyphenols of Hungarian Tokaji aszú, investigating these molecules on the inflammation-induced in vitro model system of endothelial cells.

Previous studies have demonstrated that polyphenols possess anti-inflammatory effects and vasorelaxation activity due to the suppression of the release of pro-inflammatory mediators [16,17,18]. We have also recently published several studies that investigated the relationship between the inflammation-induced HUVEC model and bioflavonoids, primarily anthocyanins [19,20]. Flavonoids are phenolic compounds which are found in plants, vegetables and fungi as secondary metabolites. Flavonoids have many health benefits, such as antioxidative, anti-inflammatory, anti-mutagenic, and anti-carcinogenic properties and can modulate key cellular enzyme functions. Furthermore, flavonoids are currently considered a potential component in a variety of nutraceutical, pharmaceutical, medicinal and cosmetic applications [21].

Given the large number of flavonoids concentrated in wine, their positive health effects and the proven beneficial effects of moderate wine consumption, we have developed an experimental system for the selective extraction as well as analytical testing of flavonoids and to investigate their biological activity in the HUVEC model.

## 2. Results

### 2.1. Analytical Studies 

Based on the results of the HPLC analyses, the main flavonoid components of HAE are as follow: Gallic acid, (+)-Catechin, Caftaric acid, Vanillic acid, (−)-Epigallocatechin gallate, Caffeic acid, Syringic acid, (−)-Epicatechin gallate, p-Coumaric acid, Ferulic acid, Ellagic acid. (+)-Catechin and Vanillic acid were represented in the extract at the highest concentration of these compounds, where caftaric acid was represented at the lowest concentration. The chromatogram in Figure 1. illustrates the results of the separation and the individual peaks that represent the individual compounds. The Table 1 shows the exact concentrations of each compound and the detection wavelengths.

### 2.2. Seeking Optimal HAE Concentration

#### 2.2.1. Results of Viability Tests

First, viability, apoptosis, necrosis, and antioxidant capacity tests were carried out to dechiper the optimal concentration of HAE. First, the effect of HAE on cell viability was investigated using an MTT-assay. The cells were exposed to HAE at different concentrations (0.05–500 μg/mL) for 24 and 48 h. We found that up to 500 μg/mL of HAE did not cause a decrease in the viability of endothelial cells (Figure 2).

To confirm the results of the MTT assay, another viability test was conducted. Intracellular lipid droplets, which are proportional to the number of living cells, were labelled with NRed fluorescent dye. The results of the NRed-assay, in line with the MTT data, showed that HAE does not exert a significant cytotoxic effect (Figure 3).

DilC_1_(5) and SYTOX Green fluorescent dyes are widely used for the investigation of apoptosis and necrosis [22]. We aimed to discover whether HAE influences the apoptotic and necrotic processes that could not be detected in the MTT assay. Consequently, the effect of HAE on the cells was examined in this regard. Based on the results shown in Figure 4 and Figure 5, HAE does not have a significant necrotic or apoptotic effect, up to 500 μg/mL. These results are consistent with those observed in the MTT and NRed assays. Overall, our results in this regard showed that HAE can be applied without the risk of any biologically relevant cytotoxic actions in this concentration range.

#### 2.2.2. The Antioxidant Effect of HAE

In order to exclude low concentrations of HAE from further experiments, we examined which concentration was capable of exerting the most significant antioxidant effect. In our experimental system, we induced an increased ROS production using H_2_O_2_. As expected, H_2_O_2_ significantly increased the production of ROS compared to the control group (Figure 6). This increment was already able to be significantly reduced by HAE with a concentration of 1 μg/mL The most marked ROS elimination was observed at concentrations of 100 and 500 μg/mL HAE. Given that, statistically, 100 μg/mL of HAE was already able to exert the strongest antioxidant effect, this concentration was chosen for further experiments.

#### 2.2.3. The Effect of HAE on Inflammatory Response in HUVEC

The basic objective of our work was to evaluate the possible positive effects of HAE on cell damage caused by LPS. In our experimental system, we first examined the mRNA expression of cytokines whose expression is specific and significantly increased during inflammation. Relative mRNA transcriptional levels of interleukin (IL)-1α, IL-1β, IL-6, and IL-8 were investigated (Figure 7). As expected, the relative mRNA expression of the investigated genes significantly increased as a result of LPS treatment compared to the control group. HAE treatment alone did not significantly influence the changes in the expression of the tested genes. These LPS-induced increments were able to be alleviated by HAE treatment in the case of all genes excluding IL-6.

#### 2.2.4. The Effect of HAE on The Expression of eNOS 

Considering the vasorelaxant effect of bioflavonoids and the relationship between the examined cytokines, inflammation and vasodilation, we examined the expression of eNOS (Figure 8), which is an excellent and essential marker for characterizing the state of vasomotor tone, at the mRNA level, in order to evaluate the effect of HAE in this regard in our in vitro model. As a result of the LPS treatment, the expression of eNOS decreased significantly. HAE was able to positively influence this change.

## 3. Discussion

We commenced our investigations with an analytical examination of the Hungarian aszú from the Tokaj wine region. After sample precondition with SPE, 11 phenolic components were detected in our aszú extract using HPLC. (+)-Catechin and Vanillic acid were represented in the extract at the highest concentration of these compounds. It is important to note that a recent study did not detect these compounds in their wine samples. As far as we can see, the reason for this contrast is the difference in the complex processes of winemaking, including multivariate interactions between grape variety, botrytis, yeast species, wine matrix and temperature effects [23]. Both compounds have been described as having potent positive health effects. Several clinical studies have noted the beneficial effects of catechin due to its antioxidant action [24]. Calixto-Campos et al. reported that vanillic acid inhibits inflammatory pain by inhibiting, among others, oxidative stress and cytokine production in mice [25].

During our in vitro experiments, we first assessed the optimal concentration of HAE. Viability tests (MTT, NRed), apoptosis and necrosis assays were performed to obtain the non-harmful concentration. The extremely strong antioxidant capacity of polyphenols is well known. Several studies have described their excellent ability to alleviate oxidative stress [26]. Numerous studies have also demonstrated the close relationship between inflammation and oxidative stress [27,28]. Based on our results, a concentration of 100 μg/mL of HAE was selected for further experiment.

The relative mRNA transcription levels of IL-1α, IL-1β, IL-6, and IL-8 were evaluated during the examination of the management possibilities of the cells for inflammation caused by LPS. Inflammation is a complex protective response of the immune system that includes the recognition of outer pathogenic structures or endogenous non-infectious molecules by different receptors [29]. The dysfunction of the endothelial cell is a key factor in the progression of a number of pathological processes [30]. Numerous studies have reported that the endothelial cells induced by inflammation mediate the enhanced production of several pro-inflammatory cytokines, including interleukin (IL)-1α and IL-1β, IL-6, and IL-8 [31,32]. IL-1α and IL-1β are equally potent inflammatory cytokines that deepen the inflammatory process, and their deregulated signalling causes cell damage [33]. The exposure of endothelial cells to LPS or TNF, routinely used inflammatory agents, results in an intensified IL-1α and IL1-β expression, which provides an important insight into the biological processes of IL-1α–IL-1R1 signalling during inflammation [34]. Similarly to the IL-1 family, IL-6 and IL-8 are also soluble mediators of inflammation and immune response. IL-6 may play a central role in host defence mechanisms because of its effect on B-lymphocyte, T-lymphocyte, and hematopoietic stem cells, among others [35]. In the case of IL-8, it has been described that increased endothelial permeability due to inflammation is associated with high expression of IL-8 at the site of damaged endothelium, which suggests that IL-8 was involved in the regulation of endothelial permeability caused by inflammation [36].

The anti-inflammatory effect of individual polyphenols has been quite well documented [37,38]. It is also well known that polyphenols have excellent antioxidant properties [39], furthermore, ROS production plays a key role in inflammation [40]. Based on the preliminary experiments presented above (Figure 6), with particular regard to the antioxidant investigations, we expected to obtain promising results from the gene expression tests. Kalló et al. recently created a comprehensive functional data set related to the identified components of aszú wine, emphasising the importance of components with potential health benefits including anti-inflammatory effects [41]. However, relatively little data are available on the anti-inflammatory effect of extracts from Hungarian aszú wines. Although many in vitro experiments have been performed with wine extracts [42], to the best of our knowledge, no research has been conducted that used an in vitro model system of endothelial cells to investigate aszú from the Tokaj wine region.

By examining the above cytokines, we aimed to evaluate the inflammatory state of the cells. Of course, we must mention that, at the same time, this is a simplified approach, considering that individual cytokines are part of an extremely complex network. Our results are consistent with the recently published research, which described that flavonoids are able to reduce the expressions of the cytokines we investigated, which are IL-1α [43], IL-1β [44] and IL-8 [45]. However, in the case of IL-6, in comparison, our results do not reflect what has been reported in the literature [46]. Several studies have described that the molecular mechanisms involved in the anti-inflammatory effect of flavonoids may include the effective inhibition of the central inflammatory players NF-kB, Nrf-2, AP-1 and mitogen-activated protein kinase (MAPK) [47]. It is well documented that the overexpression of these factors results in increased cytokine production [48,49,50]. In the case of IL-6, the unexpected change that occurred as a result of HAE treatment raises the possibility that the point of attack of the tested active ingredient is not a central factor that plays a key role in inflammation. From this point of view, it would be adequate to compare the effect of proven inhibitors of the investigated cytokines with the effect of HAE.

The cytokines presented above, due to their pleiotropic effect, play an important role in other intracellular processes in addition to inflammatory reactions, including endothelial vasorelaxation [51]. Nitric oxide (NO) produced by endothelial nitric oxide synthase (eNOS) is responsible for overall vascular homeostasis [52]. During inflammation, NO production is significantly damaged, which contributes to the deterioration of endothelial function [53]. It has been reported that flavonoids increase the expression of eNOS and decrease the expression of various genes responsible for vasoconstriction [54]. Our results, consistent with previous studies, showed that HAE can ameliorate LPS-induced damage by enhancing endothelium-dependent vasodilation through eNOS expression.

In summary, Hungarian aszú extract rich in polyphenols was prepared, and its individual components were analyzed using HPLC. HAE is able to influence intracellular eventswithout influencing the viability, necrosis or apoptosis of HUVECsin our inflammation model, including the modulation of the expression of investigated inflammatory cytokines as well as their vasoactive effects. Of course, further studies are needed to evaluate the anti-inflammatory activity in vivo in order to assess whether HAE has a usable therapeutic potential. Considering the alcohol content of wine, it is important to emphasize the strict limits of the possible therapeutic potential of HAE.

Nevertheless, this research contributes to a more thorough understanding of the bioactive molecules of aszú from the Tokaj wine region, in terms of both the bioflavonoid profile and their effect on the model system of endothelial cells. Simultaneously, our investigations fit into the context of international research related to the active ingredients in wine.

## 4. Material and Methods

### 4.1. Materials

#### Chemicals

All the reagents were obtained from the distributor of iBioTech Hungary Ltd. (iBioTech Hungary Ltd., Budapest, Hungary).

### 4.2. Methods

#### 4.2.1. Preparation of Hungarian Aszú Extract (HAE)

The sample preparation and cleaning procedure were carried out based on the work of Homoki et al. [55]. Briefly, the phenolic compounds were extracted and concentrated from a Hungarian aszú wine sample using solid phase extraction (SPE). This step was performed using vacuum manifold equipment (Waters, Milford, MA, USA). After centrifugation (10 min 4500 min^−1^), the wine sample was purified using Supelclean ENVI-18 SPE columns. The stationary phase was conditioned with 5 mL of methanol, then with 5 mL of distilled water, and finally with 1 mL of Hungarian aszú wine sample. After, the columns were washed with 5 mL of methanol containing 5% water to remove interfering components. The phase rich in polyphenol components was eluted with methanol containing 20% water. Finally, the solvent was evaporated at 40 °C with Büchi Rotavapor R-210 (BÜCHI Labortechnik AG, Flawil, Switzerland). For the in vitro experiments, the treatment solution was prepared with the cell maintenance solution (M199 medium), the concentration of which was 1 mg/mL. The dilutions were then carried out with the M199 medium.

#### 4.2.2. High-Performance Liquid Chromatography

The HAE was analyzed using the Waters Alliance e2695 separation module (Waters, Milford, MA, USA) equipped with a Waters 2998 Photodiode Array Detector (PDA). Data acquisition was performed with Waters Empower 3 software. The polyphenols of HAE were analyzed using a Hypersil ODS 250 mm × 4.6 mm column with a 5 µm particle size (Agilent Technologies, Santa Clara, CA, USA) using gradient elution. Eluent A was methanol, eluent B was 3% formic acid in water. The samples were previously filtered in a 0.22 µm PTFE syringe filter (Filter-Bio, Nantong, China) and the injection volume was 10 µL. The elution gradient was used as follows: 0 min 15% A; 15 min, 28% A; 20 min, 30% A; 25 min, 35% A; 30 min, 40% A; 33 min, 48% A; 37 min, 50% A; 40 min, 50% A; 41 min 15% A; 45 min and 15% A. PDA was applied at 280 and 320 nm. The flow rate was 1 mL min^−1^, and the oven temperature was 30 °C.

#### 4.2.3. Cell Culture

HUVEC/TERT 2 was obtained from ATCC (ATCC, Manassas, VA, USA). The cell line was isolated from the vascular endothelium of a white female patient. To culture the cells, M199 medium (Biosera, Nuaille, France) was used, which was supplemented with 10% heat-inactivated FBS, 1% penicillin/streptomycin, 1% amphotericin B, 2 mM glutamine (Biosera, Nuaille, France) and endothelial cell growth medium-2 (Lonza, Basel, Switzerland). The cells were maintained in an Esco CelCulture^®^ CO_2_ Incubator (Esco Lifesciences Group, Singapore) at 37 °C, under 5% CO_2_. Before seeding, to support the adhesion of the cells, 0.1% gelatin solution was used. To create the inflammatory model, LPS (eBioscienc, San Diego, CA, USA) was added to the M199 medium to a final concentration of 100 ng/mL. The cells were divided into four groups, 24 h of incubation with basic medium (control), 24 h incubation with 100 ng/mL of LPS (LPS), 24 h incubation with 100 μg/mL of HAE (100 μg/mL HAE) and 24 h incubation with 100 ng/mL of LPS plus 100 μg/mL of HAE (LPS + 100 μg/mL HAE).

#### 4.2.4. Determination of Cellular Viability

The endothelial cell viability was determined using a 3-[4,5-dimethylthiazol-2-yl]-2,5 diphenyltetrazolium bromide (MTT) assay. The cells were seeded in 96-well plates at a density of 20,000 cells per well and were treated with a Hungarian aszú extract (HAE) of different concentrations (0.05, 0.1, 0.5, 1, 5, 10, 50, 100 and 500 μg/mL) and without HAE (control group) for 24 and 48 h. The contents of the wells were aspirated and the cells were incubated with 100 μL of MTT solution for 3 h. The formazan crystal formed during incubation was dissolved in 100 μL of solubilizing solution (81% (*v*/*v*), isopropyl alcohol, 9% (*v*/*v*), 1 M HCl, 10% (*v*/*v*) Triton (X-100) and the absorbance was assessed at 465 nm using a Clariostar microplate reader (BMG Labtech, Ortenberg, Germany). The results were expressed relative to the control group.

#### 4.2.5. Determination of Intracellular Lipids

For the quantitative measurement of the lipid content, endothelial cells (20,000 cells/well) were cultured in 96-well plates in quadruplicate and were treated as indicated in Section 4.2.4. Subsequently, the supernatants were discarded, the cells were washed twice with phosphate-buffered saline (PBS) and 100 µL of a 1 µg/mL Nile Red solution in PBS was added to each well. The cells were incubated at 37 °C for 30 min, and fluorescence was assessed at 485 nm excitation and 565 nm using a Clariostar microplate reader. The results were expressed relative to the control group.

#### 4.2.6. Determination of Apoptosis

To assess possible apoptotic events, the mitochondrial membrane potential of the HUVECs was evaluated using 1,1′,3,3,3′,3′-hexamethyl-indodicarbocyanin iodide (DilC_1_(5)) dye. Similar to an MTT assay, the endothelial cells were plated to 96-well plates at a density of 20,000 cells per well and treated as indicated in Section 4.2.3. After the removal of the medium, the cells were incubated for 30 min with 50 μL/well DilC_1_(5) working solution (50 nM/L dissolved in Dulbecco’s modified Eagle’s medium). Subsequently, the cells were washed twice with PBS. The fluorescence was measured at 630 nm excitation and 670 nm emission wavelengths using a Clariostar microplate reader (BMG Labtech, Ortenberg, Germany). The results were expressed relative to the control group.

#### 4.2.7. Determination of Necrosis

The necrotic processes were evaluated using SYTOX green staining. The endothelial cells were cultured in 96-well plates and treated as indicated in Section 4.2.3. Then supernatants were discarded, and the cells were incubated for 30 min with 50 μL/well of SYTOX green dye (1 μM dissolved in Dulbecco’s modified Eagle’s medium). After incubation, the cells were washed with PBS. The fluorescence of SYTOX Green was measured at 490 nm excitation and 520 nm emission wavelengths using a Clariostar microplate reader (BMG Labtech, Ortenberg, Germany). The results were expressed relative to the control group.

#### 4.2.8. Evaluation of ROS Production 

The endothelial cells were seeded in a 24-well plate. To label intracellular ROS the cells were exposed to 100 μMol/L of 2′,7′-dichlorofluorescin diacetate (DCFDA) for 1 h at 37 °C. Thereafter, the cells were washed twice with PBS and treated with 100 μMol/L of H_2_O_2_. M199 medium was used as the control. The HAE of different concentrations (0.05, 0.1, 0.5, 1, 5, 10, 50, 100 and 500 μg/mL) was used to test the antioxidant effect of the extract. The level of ROS was quantified by measuring the fluorescence intensity of DCFDA, which is proportional to the production of ROS, using a Clariostar microplate reader (BMG Labtech, Ortenberg, Germany). The results were expressed relative to the control group. The sequence of the primers belonging to the studied genes is shown below.

#### 4.2.9. Quantitative Real-Time PCR (qPCR)

qPCR was implemented in a Roche LightCycler 480 System (Roche, Basel, Switzerland) using the 5′ nuclease assay. The total RNA was isolated using Extrazole. Reverse transcription was performed in a Life ECO thermal cycler (Bioer Technology Co., Ltd., Hangzhou, China) using an UltraScript 2.0 cDNA synthesis kit (PCR Biosystems, London, UK). The cDNA was amplified using the 2 × qPCRBIO Probe Mix No-ROX assay (PCR Biosystems, London, UK). The gene of *glyceraldehyde 3-phosphate dehydrogenase* was applied as an internal control. The results were expressed relative to the control group. The primers used in the qPCR tests are in Table 2.

#### 4.2.10. Statistical Analysis 

The data were analyzed and presented using GraphPad Prism 8.3.1 (GraphPad Software, La Jolla, CA, USA). For multiple comparisons, the results were analyzed using an ANOVA with Tukey’s post hoc test. The data were exhibited in the form of the mean ± SEM, excluding PCR, where the data were presented as the mean ± SD. Differences were considered statistically significant, when *p* < 0.05.

## Figures and Tables

**Figure 1 ijms-24-06124-f001:**
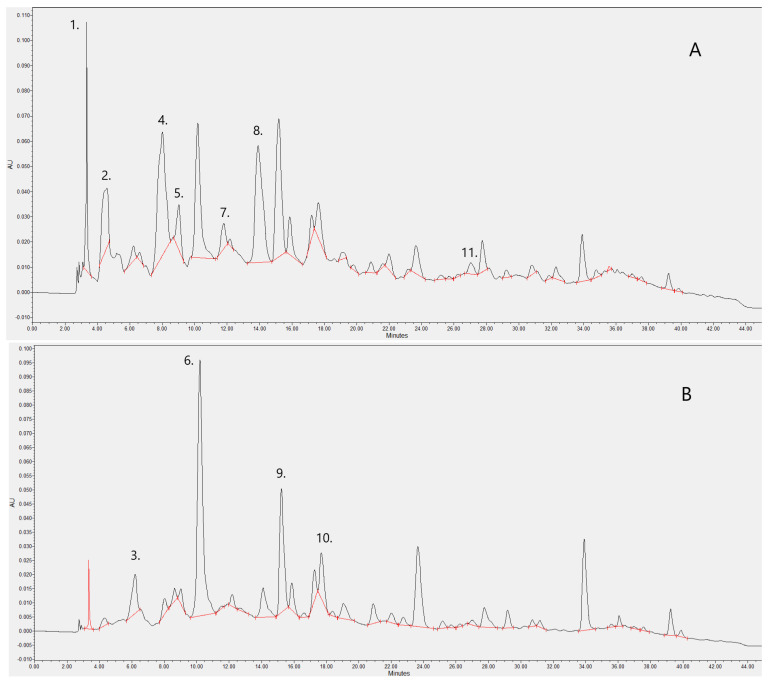
Polyphenols of HAE were analysed on a Hypersil ODS 250 mm × 4.6 mm column with 5 µm particle size using gradient elution and the compounds were detected at 280 nm (**A**) and 320 nm (**B**). The following peaks were identified on the basis of the retention time of the authentic standards: (1) Gallic acid, (2) (+)-Catechin, (3) Caftaric acid, (4) Vanillic acid, (5) (−)-Epigallocatechin gallate, (6) Caffeic acid, (7) Syringic acid, (8) (−)-Epicatechin gallate, (9) p-Coumaric acid, (10) Ferulic acid, (11) Ellagic acid.

**Figure 2 ijms-24-06124-f002:**
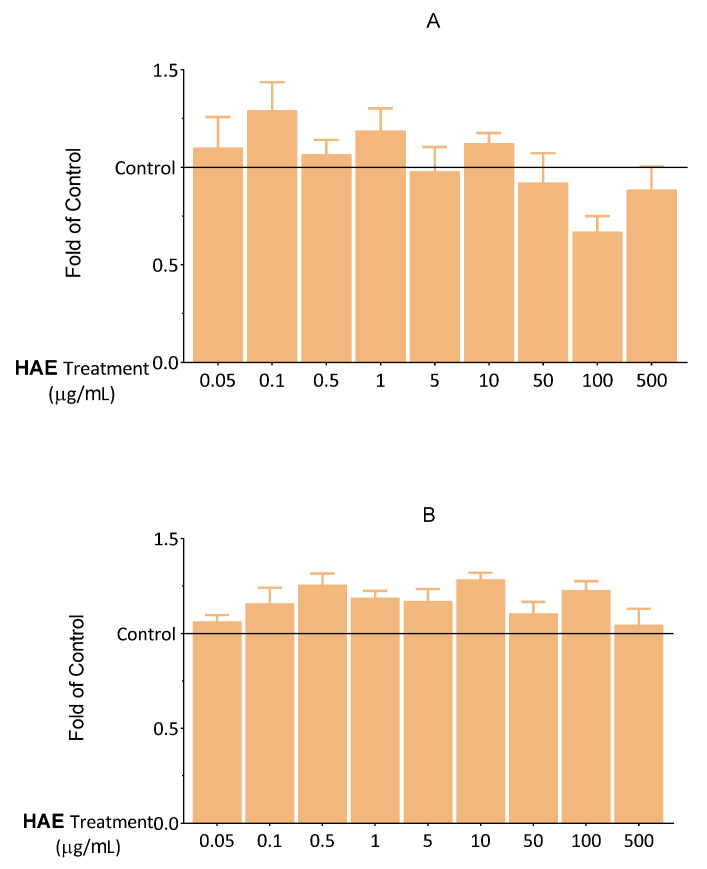
Viability of HUVECs was examined after 24 (**A**) and 48 (**B**) hours using an MTT assay. The results are expressed relative to the control (Control = 1; 0 μg/mL HAE). Data are expressed as the mean ± SEM of four individual experiments. The thin grey solid line indicates the control level (Control = 1). Differences were considered statistically significant when *p* < 0.05. We did not observe any statistically significant differences.

**Figure 3 ijms-24-06124-f003:**
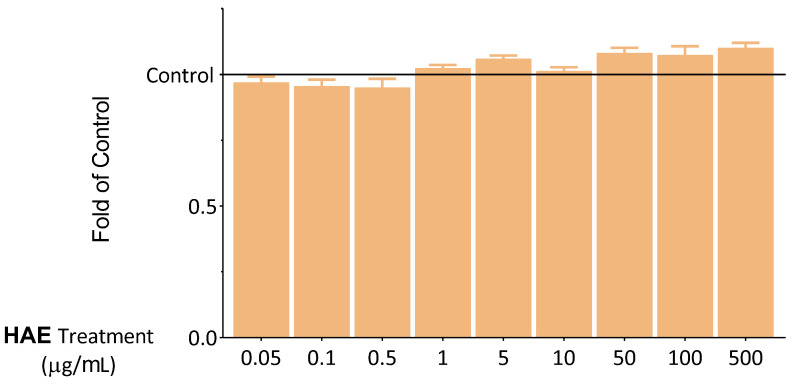
Viability of HUVECs was examined after 24 h using NRed. The results are expressed relative to the control (Control = 1; 0 μg/mL HAE). Data are expressed as the mean ± SEM of four individual experiments. The thin grey solid line indicates the control level (Control = 1). Differences were considered statistically significant, when *p* < 0.05. We did not observe any statistically significant differences.

**Figure 4 ijms-24-06124-f004:**
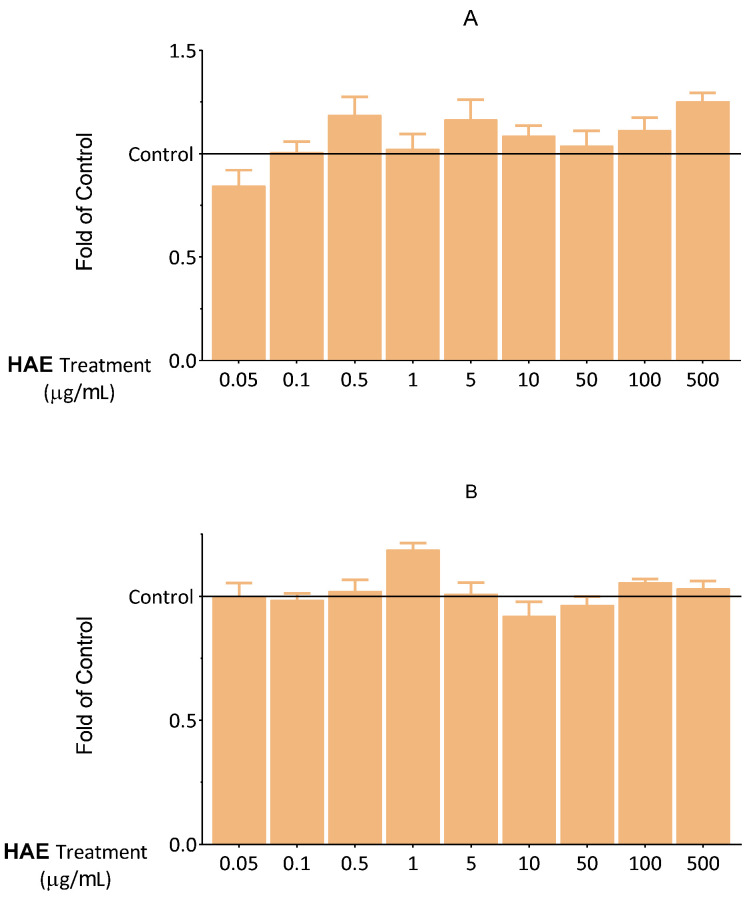
Fluorescent DilC_1_(5) labeling. The effect of HAE on apoptosis after 24 (**A**) and 48 (**B**) hours. The results are expressed relative to the control (Control = 1; 0 μg/mL HAE). Data are expressed as the mean ± SEM of four individual experiments. The thin grey solid line indicates the control level (Control = 1). Differences were considered statistically significant when *p* < 0.05. We did not observe any statistically significant differences.

**Figure 5 ijms-24-06124-f005:**
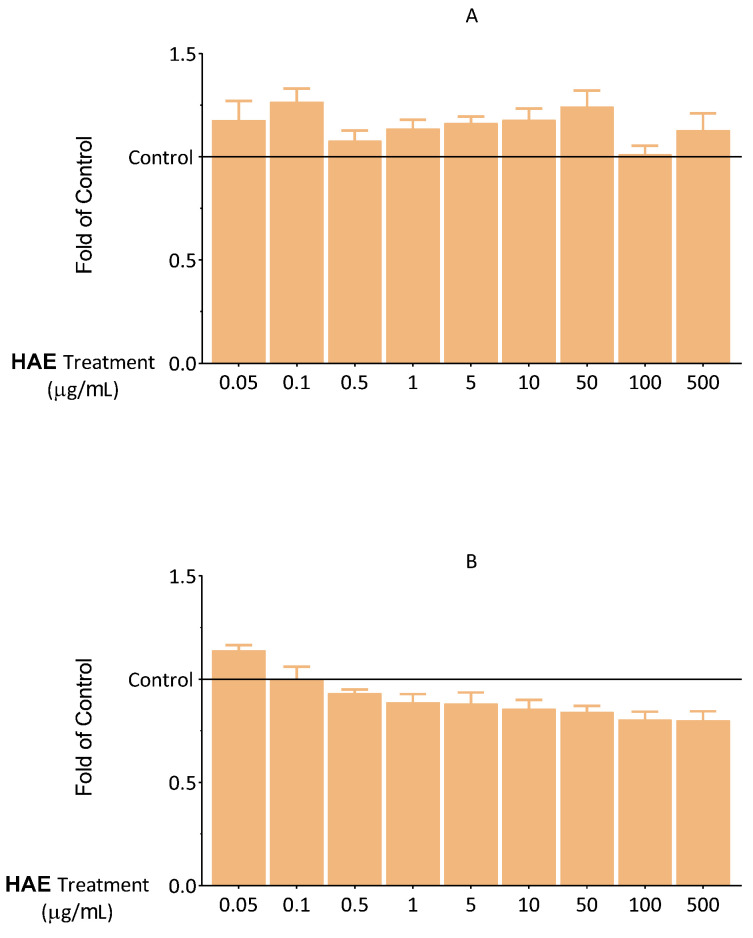
Fluorescent SYTOX green labelling. The effect of HAE on necrotic processes after 24 (**A**) and 48 (**B**) hours. The results are expressed relative to the control (Control = 1; 0 μg/mL HAE). Data are expressed as the mean ± SEM of four individual experiments. The thin grey solid line indicates the control level. Differences were considered statistically significant, when *p* < 0.05. We did not observe any statistically significant differences.

**Figure 6 ijms-24-06124-f006:**
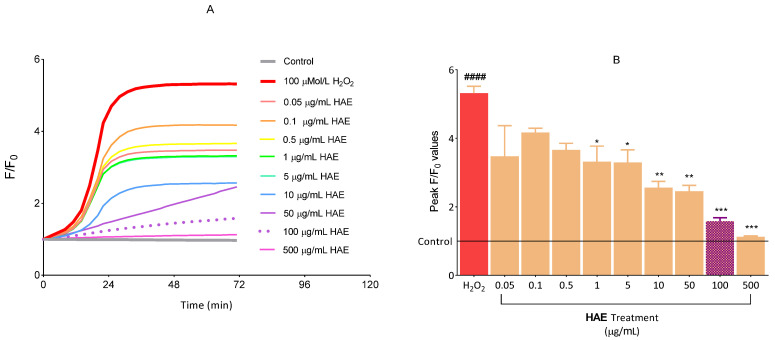
The antioxidative effect of HAE on endothelial cells incubated with or without 100 μMol/L H_2_O_2_. To label intracellular ROS the cells were exposed to 100 μMol/L DCFDA for 1 h at 37 °C. Fluorescent intensity was normalised to the baseline (**A**). Statistical analysis was performed at the peak fluorescence (F/F0) values (**B**). Data are expressed as the mean ± SEM. #### mark significant (*p* < 0.0001) differences between control and H_2_O_2_ group, * (*p* < 0.05), ** (*p* < 0.01), *** (*p* < 0.001) mark significant differences between H_2_O_2_ and H_2_O_2_ + HAE groups. H_2_O_2_: hydrogen-peroxide.

**Figure 7 ijms-24-06124-f007:**
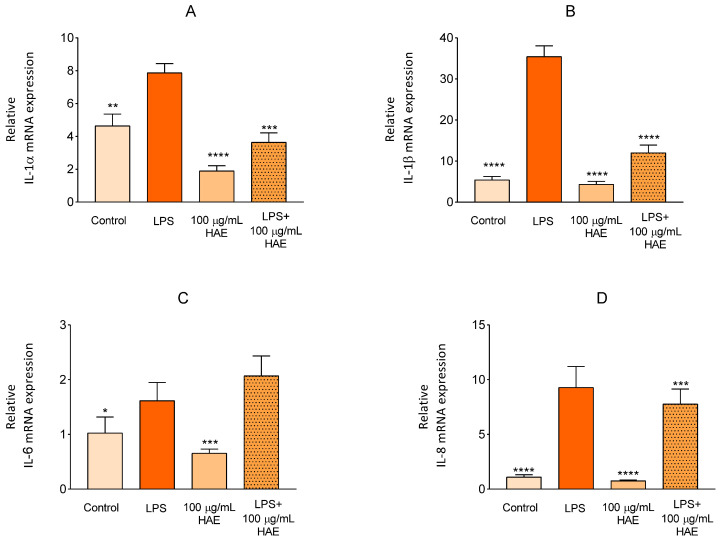
The anti-inflammatory action of HAE. qPCR analyses of gene expression of IL-1α (**A**), IL-1β (**B**), IL-6 (**C**) and IL-8 (**D**) on HUVECs following the indicated 24 h of simultaneous treatment. Total RNA was isolated by using Extrazole. Reverse transcription was performed in a Life ECO Thermal Cycler by using an UltraScript 2.0 cDNA Synthesis Kit. The cDNA was amplified by using the 2 × qPCRBIO Probe Mix No-ROX assay. Data are presented by using the ΔΔCT method regarding glyceraldehyde-3-phosphate dehydrogenase (GAPDH)-normalized mRNA expressions of the untreated control. *, **, *** and **** mark significance (*p* < 0.05, *p* < 0.005, *p* < 0.0005 and *p* < 0.0001).

**Figure 8 ijms-24-06124-f008:**
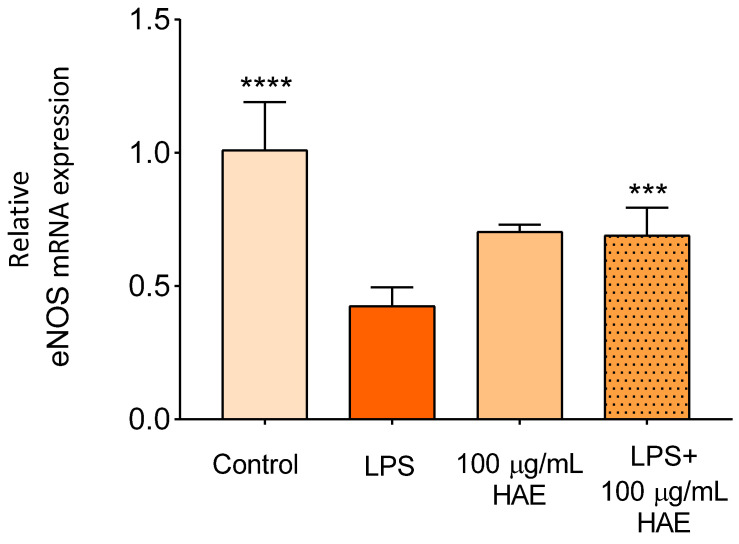
The effect of HAE on expression of eNOS. Total RNA was isolated by using Extrazole. Reverse transcription was performed in a Life ECO Thermal Cycler by using an UltraScript 2.0 cDNA Synthesis Kit. The cDNA was amplified by using the 2 × qPCRBIO Probe Mix No-ROX assay. Data are presented using the ΔΔCT method regarding glyceraldehyde-3-phosphate dehydrogenase (GAPDH)-normalised mRNA expressions of the untreated control. *** and **** mark significant (*p* < 0.0005 and *p* < 0.0001).

**Table 1 ijms-24-06124-t001:** Flavonoid compounds detected in HAE, their compound families, exact concentrations and detection wavelengths.

Number	Phenolic Compound	Compound Family	λ Detection (nm)	Concentration (mg/L)
1	Gallic acid	Hydroxybenzoic acid	280	0.843
2	(+)-Catechin	Flavan-3-ol	280	9.963
3	Caftaric acid	Hydroxycinnamic acid	320	0.268
4	Vanillic acid	Hydroxybenzoic acid	280	9.262
5	(−)-Epigallocatechin gallate	Flavan-3-ol	280	0.946
6	Caffeic acid	Hydroxycinnamic acid	320	1.768
7	Syringic acid	Hydroxybenzoic acid	280	0.355
8	(−)-Epicatechin gallate	Flavan-3-ol	280	4.426
9	p-Coumaric acid	Hydroxybenzoic acid	320	0.638
10	Ferulic acid	Hydroxycinnamic acid	320	0.277
11	Ellagic acid	Hydroxybenzoic acid	280	0.417

**Table 2 ijms-24-06124-t002:** Primers used in qPCR studies.

IL-1α	Forward	GCA TTA CAT AAT CTG GAT GAA GCA G
	Reversed	GGT TTT GGG TAT CTC AGG CAT C
	Probe	6-Fam-TGG TCT TCA TCT TGG GCA GTC ACA-BHQ-1
IL-1β	Forward	GGC AAT GAG GAT GAC TTG TTC
	Reversed	CGG AGA TTC GTA GCT GGA TG
	Probe	6-Fam-TGA TGG CCC TAA ACA GAT GAA GTG-BHQ-1
IL-6	Forward	AAT TCG GTA CAT CCT CGA CG
	Reversed	GAT TTT CAC CAG GCA AGT CTC
	Probe	6-Fam-TGT TAC ATG TCT CCT TTC TCA GGG C-BHQ-1
IL-8	Forward	TCC TGA TTT CTG CAG CTC TG
	Reversed	GTC CAC TCT CAA TCA CTC TCA G
	Probe	6-Fam-CAT ACT CCA AAC CTT TCC ACC CC-BHQ-1
eNOS	Forward	CTT CCT GGA CAT CAC CTC C
	Reversed	AAC CAC TTC CAC TCC TCG
	Probe	6-Fam-TCC TGC TGT TCC CTG GGC-BHQ-1
GAPDH	Forward	CCT CCA CCT TTG ACG CTG
	Reversed	CTC TTC CTC TTG TGC TCT TGC
	Probe	Hex-CAT TGC CCT CAA CGA CCA CTT T-BHQ-1

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
