# Peer review of "HPLC Analysis of Polyphenols Derived from Hungarian Aszú from Tokaj Wine Region and Its Effect on Inflammation in an In Vitro Model System of Endothelial Cells"

_ijms, 2023, doi:10.3390/ijms24076124_

Round 1
Reviewer 1 Report (Previous Reviewer 1)
The authors addressed most of the reviewers' critiques, and the overall manuscript offers a preliminary proof of concept that must be thoroughly tested in future publications.
Author Response
Dear Reviewer!
We would like to thank you again for your comments and your work to improve the quality of the manuscript. We thank you also for drawing attention to the need to improve English language and style. Accordingly, we will use MDPI's "Language Editing Services".
Reviewer 2 Report (Previous Reviewer 2)
Based on the strong claims made in the specifications of the extract compounds (Table 2), I personally cannot accept what these substances are by doing just one measurement (HPLC) without performing mass spectrometry, if the editor has no problem with this, this article can be accepted.
Author Response
Dear Reviewer!
We respectfully accept that the result of the HPLC measurement without mass spectrometry does not provide sufficient evidence for you. However, we trust in the positive evaluation of our manuscript.
Finally, we would like to thank you again for your comments and your work to improve the quality of the manuscript.
Reviewer 3 Report (Previous Reviewer 3)
Dear authors
The authors re-submitted an original article, entitled "HPLC analysis of polyphenols derived from Hungarian aszú from Tokaj wine region and its effect on inflammation in an in vitro model system of endothelial cells," to the journal "International Journal of Molecular Sciences." This article was based on in vitro experiments to describe that the the bioactive molecules of Hungarian aszú can alleviate the LPS-induced endothelial dysfunction presumably due to its potent antioxidant and anti-inflammatory effects. This study is primarily based on limited data from Hungarian aszú treating the HUVEC cell line without in vivo evidence. Either, there is lack of novel findings, such as specific compound/molecule and novel extraction method to enhance the bioactivity. The re-submission didn't have more data but fine modification of descriptions. Additional in vitro experiments were essential to further elucidate the importance. The following are my comments:
1. The in vitro experiments such as human tissues or animal experiments to further elucidate the extract's antioxidant and anti-inflammatory effects are essential.
2. In Figure 2, the viability of HUVECs after treating 24 hours was much lower at 50,100,500 ug/mL compared to the control. How can the authors explain there was no cytotoxicity?
3. In table 2, (+)-Catechin, Vanillic acid, and (-)-Epicatechin gallate are top three abundant flavonoid compounds detected in the extract. In the literature, there were published articles regarding to their impacts on endothelial regulation. The authors should perform specific in vitro and in vivo studies to validate these compounds' effects on endothelial cells.
4. The mechanisms and pathways for antioxidant and anti-inflammatory effects of Hungarian aszú on HUVEC cells are not clarified or further explored.
Author Response
Dear reviewer!
Our answers are attached below.

Reviewer 4 Report (New Reviewer)
1. Please supplement the detailed experimental parameters in Figure 1-8.
2. The format of references should be unified. Please check the start and end page.
Author Response
Dear Reviewer!
Thank you for your comments and helpful advices. We have revised our manuscript and we made corrections according to your recommendations.
- As you suggested, we supplemented the figure captions in our manuscript for a more accurate description.
- References were managed with the EndNote reference manager. Nevertheless, there are indeed stylistic differences in the individual references. References have been checked to ensure they are consistent.
Round 2
Reviewer 3 Report (Previous Reviewer 3)
Dear authors
The authors re-submitted an original article, entitled "HPLC analysis of polyphenols derived from Hungarian aszú from Tokaj wine region and its effect on inflammation in an in vitro model system of endothelial cells," to the journal "International Journal of Molecular Sciences." This article was based on in vitro experiments to describe that the the bioactive molecules of Hungarian aszú can alleviate the LPS-induced endothelial dysfunction presumably due to its potent antioxidant and anti-inflammatory effects. This study is primarily based on limited data from Hungarian aszú treating the HUVEC cell line without in vivo evidence. The re-submission didn't have more data but only provide fine modification of descriptions. Since the authors seem not perform further experiments, no further comments are rendered.
This manuscript is a resubmission of an earlier submission. The following is a list of the peer review reports and author responses from that submission.
Round 1
Reviewer 1 Report
The manuscript by Markovics et al. focuses on investigating various biological effects of Hungarian Aszú Extract (HAE) on HUVEC cells in vitro and in the context of lipopolysaccharide (LPS) stimulation.
The HPLC results show that HAE is composed of eleven different flavonoids. In vitro studies showed that increasing doses of HAE did not cause a detrimental effect on cellular viability and did not induce apoptosis or necrosis. Further, HAE showed an antioxidant effect on H2O2-treated HUVEC cells in a dose-dependent manner. Finally, the authors tested the effect of HAE on the RNA expression of IL1a, IL1b, IL-6, IL-8, and eNOS in HUVEC cells treated with LPS. The manuscript is well written, and the results provide interesting proof of concept data regarding the HAE composition and some of its potential biological effects using an in vitro system.
Major comments:
- The components of HAE with a higher concentration were (+)-Catechin and vanillic acid. Is there any evidence in the literature regarding these flavonoids' potential antioxidant or anti-inflammatory effects? If so, the authors should include this in their discussion.
- Line 91: Can the authors provide more information about the wine samples used? Is there a standardized method to produce this wine? The referenced work by Kalló et al. shows that although some compounds are common among aszú wine sources, the concentration of some molecules is winery dependent.
- In the study by Kalló et al., the authors did not report the presence of (+)-Catechin or vanillic acid by their LC-MS/MS analysis as it was found in this manuscript; however, both studies report the presence of caffeic acid, among other compounds. How can the authors explain these results? This should be included in their discussion.
- What was the diluent used to solubilize and dilute the HAE? Is this reagent different from the vehicle used for HAE extraction from wine? Was this diluent used as the control the authors used in all the experiments that they refer to as "without HAE"?
- In the methods, the authors state that they run some experiments in quadruplicates, but in the figure legends, they report four individual experiments. Can the authors explain how the in vitro experiments were designed to ensure reproducibility?
- The findings that increasing concentrations of HAE up to 10,000 times more did not affect cell viability or induced apoptosis or necrosis are surprising. Is this effect comparable to other wine extracts or flavonoids (i.e., doses of 100 or 500 ug/mL)? What dose range is where cellular toxicity is usually observed with other compounds or preparations?
- As in the previous comment, what is the usual dose range where other polyphenols or wine extracts can exert antioxidant effects in vitro? How does HAE compare to them?
- Did the authors use positive controls in their MTT, NRed, DilC1(5), and SYTOX experiments as they showed for the ROS production?
- The findings about reduced RNA expression in HUVEC cells treated with LPS+HAE are exciting proof of concept. Can the authors speculate about the potential mechanisms involved in the reduced expression of pro-inflammatory cytokines induced in HUVEC cells upon treatment with LPS+HAE? Would they be similar to what has been shown in other wine extracts or flavonoids?
- The authors focused on the measurement of pro-inflammatory cytokines after LPS stimulation. Is there any evidence that flavonoids or wine extracts can induce or enhance the expression of anti-inflammatory cytokines or molecules?
Minor comments:
- Line 28: the authors state that HAE “could alleviate LPS-induced endothelial dysfunction.”
Since the authors only measured four cytokines and eNOS RNA expression, I recommend using a more nuanced and accurate term. - In the Methods section, the information about the type of LPS used and the concentration used is missing.
- Line 287: The statement "comprehensive picture of the inflammatory state of the cells" seems exaggerated and should be pondered in the actual experiments, considering that the authors only measured the RNA expression of four cytokines.
Reviewer 2 Report
In my view, this study should be re-submitted after correcting many parts and/or rejected.
Regarding the method, it is very important to provide the wine-extract extraction method. Limiting contents to one reference does not give the reader any clues. We don't really know in what solution the extract was extracted, what its solubility was, and how was the stability of the extract for a period of at least 48 hours.
The Red Nile assay for measuring fatty acids is spotty and their correlation with MTT data is far from expected (lines 203 to 206). Please modify this section or enter a ref. that has previously linked these two parameters.
In the HPLC section, some may recognize that PDA means the same detector, but its full name should be provided for other readers. By the way, how did you determine that you need to adjust the PDA at 280 and 330 nm? Please add the absorption spectrum of the extract from 250 to 600.
What solution was used to prepare the extract stock and then prepare different dilutions? In what solution was the stock prepared?
MTT is for cell viability, not for determining apoptosis. This section and related data in the method, results and discussion should be corrected.
The amount of ROS was determined by fluorescence absorption of what substance?
Where are the primers for the RT-PCR section stated?
Table 1: You can never find out the nature of the substance and even its concentrations with an rp-HPLC. These data are purely hypothetical. To determine the exact amount and its nature, mass spectroscopy should be done. If you load another substance such as vitamin A in the column, you will find that it will leave the column sometime during 44 minutes, but it cannot be concluded that vitamin A is also present in the extract. To determine it accurately, fractionation should be done and after purification of the peaks, MS should be added.
Reviewer 3 Report
Dear authors
The authors kindly submitted an original article, entitled "HPLC analysis of polyphenols derived from Hungarian aszú wine and its effect on inflammation in an in vitro model system of endothelial cells ," to the journal "International Journal of Molecular Sciences." This article was based on in vitro experiment to describe that the the bioactive molecules of Hungarian aszú could alleviate the LPS-induced endothelial dysfunction presumably due to its potent antioxidant and anti-inflammatory effect. To be honest, this study is primarily based on limited data and evidence from cell line studies without any in vivo evidence. Either, there is lack of novel findings, such as any specific compound/molecule or a new extraction method first proposed in this article to enhance the bioactivity. Readers want to have more insights from your study. Several major and minor parts were essential to be corrected or further improved. The following are my comments: 1. In the abstract, " HPLC", " qPCR", and "MTT-" should be fully described first. 2. In the abstract, the abbreviation of LPS was not well indexed at its first appearance. 3. In the abstract, authors said they evaluated bioactive molecules of "Hungarian aszú", but didn't clearly and specifically said the wine is Tokaji Aszú. So, the authors should make it clear and then use "aszu from the Tokaj wine region" in their last sentence. 4. In the introduction, the authors should well introduce their main investigated source, Hungarian Aszú. They should explain this wine's characteristics, related studies, special components, and possible target biomolecules. But, none was seen. 5. In the session 2.2.1. Preparation of Hungarian Aszú Extract (HAE), the authors should at least briefly describe the extraction process, how many times they did, and to what extent they qualified and accepted. 6. In the session 2.2.2, HPLC and PDA should be open spelled first. 7. In the session 2.2.4, "MTT (3-[4,5-dimethylthiazol-2-yl]-2,5 diphenylte- 117 trazolium bromide)" the abbreviation should be after the complete description. 8. In the session 2.2.5, PBS should be open spelled first. 9. In the sessions, 2.2.7. Determination of necrosis and 2.2.8. Evaluation of ROS production, "Cells were cultured in 96-well plates.""The cells were seeded in a 24-well plate." The authors should clearly explain what cells were at the beginning. 10. In the session 2.2.10. Statistical analysis, the authors should describe the software they use and the p<0,05 is one-tailed or two-tailed. 11. The authors used only one human HUVEC/TERT 2 cell line to valid the Hungarian aszú extract's cellular viability, apoptosis, and necrosis. More human cell lines are needed for reducing the single cell line bias. 10. In the materials and method, the LPS-induced endothelial dysfunction was not introduced. 11. No specific biomolecule or new compound was further analyzed and validated separately from Hungarian aszú extract in this study, although the authors did HPLC. 12. Except flavonoid compounds, the authors should provide other compounds' data they analyzed from the wine as a supplementary table. 13. No data from animal experiments or human samples were provided for further validation. 14. Results and discussion should be separated. The combination made the discussion not integrative but fragmented.